# Photosynthetic Pigment and Carbohydrate Profiling of *Fucus vesiculosus* from an Iberian Coastal Lagoon

**DOI:** 10.3390/plants12061324

**Published:** 2023-03-15

**Authors:** Ana C. R. Resende, Rui Pereira, Cláudia Nunes, Sónia Cruz, Ricardo Calado, Paulo Cartaxana

**Affiliations:** 1Laboratory for Innovation and Sustainability of Marine Biological Resources (ECOMARE), Centre for Environmental and Marine Studies (CESAM), Department of Chemistry, University of Aveiro, 3810-193 Aveiro, Portugal; anacrresende@ua.pt; 2A4F—Algae for Future Lda., Campus do Lumiar, Estrada do Paço do Lumiar, Edifício E R/C, 1649-038 Lisboa, Portugal; rui.pereira@a4f.pt; 3Aveiro Institute of Materials (CICECO), Department of Materials and Ceramic Engineering, University of Aveiro, 3810-193 Aveiro, Portugal; claudianunes@ua.pt; 4ECOMARE, CESAM, Department of Biology, University of Aveiro, 3810-193 Aveiro, Portugal; sonia.cruz@ua.pt (S.C.); rjcalado@ua.pt (R.C.)

**Keywords:** fucoidans, fucoxanthin, macroalgae, monosaccharides, photosystem II

## Abstract

*Fucus vesiculosus* is a brown seaweed with applications in the food, pharmaceutic, and cosmetic industries. Among its most valuable bioactive compounds are the pigment fucoxanthin and polysaccharides (e.g., fucoidans). In this study, we profiled the photosynthetic pigments and carbohydrates of *F. vesiculosus* from six locations along the Ílhavo Channel in the Iberian coastal lagoon of Ria de Aveiro, Portugal. Photosynthetic performance (Fv/Fm), pigment, and carbohydrate concentrations were similar between locations, despite differences in environmental factors, such as salinity and periods of exposure to desiccation. Concentration of total carbohydrates (neutral sugars + uronic acids) averaged 418 mg g^−1^ dw. Fucose was the second most abundant neutral sugar, with an average concentration of 60.7 mg g^−1^ dw, indicating a high content of fucoidans. Photosynthetic pigments included chlorophylls *a* and *c*, β,β-carotene, and the xanthophylls fucoxanthin, violaxanthin, antheraxanthin, and zeaxanthin. Concentrations of fucoxanthin were higher than those reported for most brown macroalgae, averaging 0.58 mg g^−1^ dw (65% of total carotenoids). This study indicates that *F. vesiculosus* from Ria de Aveiro is a valuable macroalgal resource for aquaculture companies operating in the region, with considerable potential to yield high-value bioactive compounds.

## 1. Introduction

Algae may play an important role in the development of innovative, healthy and sustainable food systems, contributing to a sustainable and circular bioeconomy [1]. In addition to their contribution to primary production, uptake of dissolved nutrients from the environment, coastal defences, and carbon sequestration, algae are used in several commercial applications as raw materials for feed and fodder, fertilizers, biofuels, bioremediation, biomonitoring, and biopolymer production, as well as in the cosmetic and pharmaceutical industries [2]. The largest volume of macroalgae production in the EU still originates from wild harvesting, although several seaweed species play a structural role in coastal ecosystems [3,4]. Hence, one of the main goals of the EU algae sector is to increase the sustainable production of macroalgae [1].

Macroalgae have gained additional attention due to their various health-promoting bioactive compounds [5,6]. The content of these high-value compounds varies between taxa and is affected by seasonality, geographical distribution, and environmental factors, such as temperature or light. The bladder wrack *Fucus vesiculosus* Linnaeus 1753 is a large brown seaweed characterized by its frond with a prominent midrib; oval to spherical air bladders, usually paired; and dichotomously branched. This species is similar to *Fucus spiralis* Linnaeus 1753 with which it hybridizes. *Fucus vesiculosus* has a wide distribution, dominating specific shallow intertidal macroalgal communities on rocky shores. It is the most studied species of the genus *Fucus* with several benefits to human health, namely suppressing mineral deficiencies, displaying high iodine content and bioactive compounds, such as fucoidans and fucoxanthin [7,8,9]. Fucoidans, mainly derived from brown seaweeds, are a type of polysaccharide that contains substantial percentages of L-fucose and sulphate ester groups, also named as fucose-containing sulfated polysaccharides [10,11,12]. These polysaccharides are particularly abundant in *Fucus* species and have been shown to display antitumor, anti-bacterial, anti-viral, anti-coagulant, and antioxidant properties, although their bioactivities depend on the extent and position of sulphation [9,10,11,13,14,15,16]. The photosynthetic pigment fucoxanthin is a light-harvesting carotenoid particularly abundant in brown seaweeds [17]. This xanthophyll has been shown to have anti-inflammatory, anticoagulant, immunostimulant, antioxidant, and antiobesity properties, and has been used as a colorant in several industrial products [18,19,20].

To optimize sustainable aquaculture of this brown seaweed, basic knowledge on the biochemical profiles of natural populations is required. The main objective of this study was to perform the biochemical characterization of *Fucus vesiculosus* along the Ílhavo channel, Ria de Aveiro, Portugal, as this species holds the potential to be used in aquaculture by companies operating in this region. This study focuses on photosynthetic pigment and carbohydrate composition of *F. vesiculosus*, with emphasis on the high-value compounds fucoxanthin and fucoidans.

## 2. Materials and Methods

### 2.1. Macroalgae Sampling

*Fucus vesiculosus* (Ochrophyta, Phaeophyceae) was collected from six locations from the inlet of Ria de Aveiro and along the Ílhavo Channel: Port of Aveiro (40°38′49.7″ N 8°44′04.2″ W), Marginal (40°38′42.0″ N 8°41′52.3″ W), A25 (40°37′57.2″ N 8°41′06.5″ W), Gafanha de Aquém (40°36′31.4″ N 8°41′02.3″ W), Vista Alegre (40°35′16.5″ N 8°41′06.7″ W), and N109 (40°34′27.1″ N 8°40′47.7″ W) (Figure 1).

The Ílhavo Channel is around 15 km long and ranges from 60 to 500 m in width. It is considered one of the smallest and narrowest channels of the Ria de Aveiro lagoon, receiving the freshwater input from the Boco River [21]. The salinity and the water temperature range from 0 to 38, and from 12 to 23 °C, respectively, depending on the season and distance from the mouth [21,22]. The sampling locations were chosen accordingly to a known salinity and tidal gradient: salinity increases from N109 towards Port of Aveiro, while exposure to desiccation periods increases in the inverse direction [21,22]. All samples were collected in a rocky substrate in dense populations, except for Vista Alegre that presents a muddy substrate and a low-density population. The Port of Aveiro, Marginal, and A25 locations are more influenced by anthropogenic activities, such as those occurring in the main commercial port of Aveiro and the industrial fishing port.

Sampling was performed on 15 June 2022 (summer, dry season), during low tide and in the intertidal zone, with 5 replicates being collected per sampling location. Sampling was performed by cutting the thallus 200 mm above the fixation point (Figure 2a,b). Samples were transferred to labelled plastic bags and transported to the laboratory within 2 h. Chlorophyll *a* fluorescence was assessed in vegetative blades and receptacles (reproductive tissue) immediately upon arrival (see below). The macroalgae biomass was frozen, freeze-dried, grounded, and stored at −20 °C until carbohydrate and photosynthetic pigment analyses were performed.

Carbohydrate and photosynthetic pigment concentrations were analysed for the whole macroalgae since separating vegetative and reproductive tissues is not feasible in an industrial scenario due to logistic and time-consuming constraints.

### 2.2. Chlorophyll a Fluorescence

Chlorophyll (Chl) *a* fluorescence was assessed in the vegetative and reproductive tissues of *F. vesiculosus* using a Junior-PAM (Pulse Amplitude Modulated) fluorometer (Walz, Effeltrich, Germany). The maximum quantum yield of photosystem (PS) II (Fv/Fm) was determined by calculating (Fm–Fo)/Fm, where Fm and Fo are the maximum and the minimum fluorescence of 45 min dark-adapted samples, respectively [23]. The parameter Fv/Fm was used as an indicator of the macroalgal photosynthetic performance.

### 2.3. Carbohydrate Composition

Neutral sugars were analysed as their alditol acetates and determined by gas chromatography with flame-ionization detection (GC-FID) after acid hydrolysis, and followed by reduction and acetylation using 2-deoxyglucose as the internal standard [24,25]. Dried samples were hydrolyzed with 0.2 mL of 72% H_2_SO_4_ for 3 h at room temperature, followed by 2.5 h in 1 M H_2_SO_4_ at 100 °C. Neutral sugars were converted into their alditol acetates after reduction with NaBH_4_ (15% (*w*/*v*) in 3 M NH_3_) at 30 °C for 1 h and acetylation with acetic anhydride in the presence of 1-methyl imidazole at 30 °C for 30 min. For uronic acid analysis, a hydrolysis with 1 M H_2_SO_4_ at 100 °C for 1 h was performed, and afterward the *m*-phenylphenol colorimetric assay was used [15]. The concentration of neutral sugars and uronic acids was expressed per dry weight (mg g dw^−1^). Fucose content is known to provide a broad estimate of levels of fucoidans [26].

### 2.4. Photosynthetic Pigment Analysis

Freeze-dried macroalgal samples were extracted in 95% cold-buffered methanol with 2% ammonium acetate [27]. Samples were sonicated for 1 min and transferred to −20 °C for 30 min in the dark. Extracts were filtered through 0.2 μm PTFE membrane filters and immediately injected into a Prominence i–LC 2030C HPLC system (Shimadzu, Kyoto, Japan). Chromatographic separation was carried out using a Supelcosil C18 column (250 mm length, 4.6 mm diameter, 5 μm particles; Sigma-Aldrich, St. Louis, MO, USA) and a 35 min elution program. The solvent gradient followed Kraay et al. [28], with an injection volume of 50 μL and a flow rate of 0.6 mL min^−1^. Pigments were identified from absorbance spectra and retention times and concentrations calculated from the signals in the photodiode array detector. Calibration curves were constructed with pure crystalline standards from DHI (Hørsolm, Denmark). Pigments were expressed in mg g dw^−1^.

### 2.5. Statistical Analysis

Statistical analyses were performed using RStudio (2022.02.3 + 492). The comparison between independent groups (locations) was performed either by applying ANOVA or Kruskal–Wallis (non-parametric version). A previous assessment of data normality (Shapiro–Wilk’s test) and homogeneity of variances between locations (Levene’s test) was required. Hence, ANOVA was performed if both assumptions were validated, otherwise, Kruskal–Wallis was applied. This approach was used to compare the maximum quantum yield of PSII (Fv/Fm) on algal parts (vegetative and reproductive), as well as the neutral sugars, uronic acids, and photosynthetic pigment concentrations of the whole algal structure between locations. Whenever a significant result (*p*-value < 0.05) was observed, pairwise comparisons were computed using Students’ *t*-test for independent samples or Dunn’s test in parametric or non-parametric approaches, respectively. Furthermore, to compare the Fv/Fm between algal parts in each location, individual Student’s *t*-tests for independent samples were performed. All *p*-values obtained by multiple comparisons were adjusted using Bonferroni’s correction.

## 3. Results

Analysis of the maximum quantum yield (Fv/Fm) of photosystem II varied between 0.63 and 0.78 in all samples (Figure 3; Appendix A). Photosynthetic performance was not significantly different between vegetative blades and reproductive tissues (receptacles). Significantly lower Fv/Fm was observed in the vegetative tissues of Marginal when compared to A25 (Kruskal–Wallis: H = 13.7353, df = 5, *p*-value = 0.02; Dunn test: *p*-value = 0.009), while no significant differences were detected in reproductive tissues between locations.

The total carbohydrate content (neutral sugars + uronic acids) varied between 34% and 51% of total dry-weight biomass. Neutral sugar analysis identified eight monosaccharides: mannose, fucose, glucose, galactose, xylose, rhamnose, ribose, and arabinose. Mannose, fucose, glucose, and galactose were the most abundant in all six locations, comprising 96% of total neutral sugars (Figure 4a,b). Average fucose concentrations were 60.7 mg g^−1^ dw, representing 27% of total neutral sugars in *F. vesiculosus*. Significant differences in neutral sugar concentrations between locations were restricted to the less abundant arabinose (H = 14.5045, df = 5, *p*-value = 0.01), and xylose (F = 5.855 (5,24), *p*-value = 0.001). Arabinose content was significantly higher in *F. vesiculosus* from Gafanha de Aquém and Marginal than from A25 (*p*-value = 0.01). Macroalgae from Gafanha de Aquém had significantly higher xylose content than from A25 (*p*-value = 0.0004), Port of Aveiro (*p*-value = 0.04), and N109 (*p*-value = 0.005). No significant differences between locations were observed for the uronic acids content (Figure 4a), neither for total carbohydrates content (neutral sugars + uronic acids).

Total pigments concentration ranged between 1.8 and 3.8 mg g^−1^ dw. The macroalga *F. vesiculosus* yielded seven different photosynthetic pigments: chlorophylls *a* and *c*, and the carotenoids fucoxanthin, violaxanthin, β,β-carotene, antheraxanthin, and zeaxanthin (Figure 5a,b). All locations presented a similar pattern of photosynthetic pigment content, with chlorophyll *a* and fucoxanthin being the most abundant, and antheraxanthin and zeaxanthin the less abundant. Average fucoxanthin concentrations were 0.58 mg g^−1^ dw, representing 65% of total carotenoids in *F. vesiculosus*. Zeaxanthin was the only pigment that presented significantly differences between locations (H = 15.085, df = 5, *p*-value = 0.01), with higher content on macroalgae from Port of Aveiro than from Vista Alegre and N109 (*p*-value = 0.01, *p*-value = 0.03, respectively).

## 4. Discussion

The range of photosynthetic performance observed in this study for *F. vesiculosus* (between 0.63 and 0.78) is similar to that previously reported for this brown macroalga [29,30,31]. In the brown seaweeds *Ascoseira mirabilis* and *Cystosphaera jacquinotii* higher Fv/Fm values have been observed in reproductive tissues compared to vegetative parts [32]. However, in our study, similar photosynthetic performances were registered in vegetative blades and reproductive tissues of *F. vesiculosus*. Also, similar Fv/Fm values were observed for specimens from different locations along the Ílhavo Channel of Ria de Aveiro, despite differences in salinity and exposure to periods of desiccation between the sampling sites [21,22]. Freshwater input from the Boco River, particularly during the winter, is more significant relative to the tidal flow in locations away from the Ria de Aveiro inlet [21]. Short or long-term changes in salinity levels have been shown to affect *F. vesiculosus*, with lower salinities negatively affecting photosynthesis [33,34,35]. It is important to stress that our study provides a snapshot and does not consider seasonal variations. Hence, differences in photosynthetic activity of the sampled populations during the winter cannot be ruled out, as differences in salinity levels between locations would be more pronounced.

Populations of *F. vesiculosus* from the non-tidal Baltic Sea can be particularly sensitive to desiccation during emersion periods [31]. On the other hand, populations highly adapted to the intertidal environment, such as the ones sampled in our study, can benefit from emersion. Meichssner et al. [36] found that periodical desiccation decreased the abundance and size of foulers, favouring the growth of *F. vesiculosus* in cultivation tanks. The same authors concluded that regular desiccation periods are effective for the production of clean *F. vesiculosus* in aquaculture, contributing to biomass valorisation [36]. Hence, it is reasonable to consider that regular emersion periods can benefit natural populations of *F. vesiculosus* adapted to the intertidal environment.

The resistance to desiccation of *F. vesiculosus* and other intertidal brown algae is related to the presence of fucoidans, which provide cell wall stability and antioxidant activity [7,37]. In a review on *Fucus* spp., Catarino and colleagues [7] reported that levels of fucoidans could reach up to 25% of dry weight in *F. vesiculosus*. In the latter study, total carbohydrate levels were shown to vary between 34 and 66% of total dry weight, in line with the content reported in our study. From the eight neutral sugars identified in the algal extracts, six of them are known to be present in the fucoidans of *F. vesiculosus*: fucose, glucose, galactose, xylose, mannose, and rhamnose [26,38]. The presence of uronic acids in the fucoidan structure is also reported, namely glucuronic acid [15]. However, most of the uronic acids determined in this study should be due to the presence of alginate, since brown algae are also rich in this polysaccharide [7]. Fucoidans from the genus *Fucus* tend to be fucose-rich (more than 70%), although reports diverge, and important proportions of other monosaccharides appear in some cases [26]. Fucose was the second most abundant monosaccharide, with concentrations ranging from 42 to 84 mg g^−1^ dw (27% of total neutral sugars), suggesting a significant contribution of fucoidans to total polysaccharide content. Similar fucose concentrations (51–116 mg g^−1^ dw) have been found in *F. vesiculosus* from shallow waters of the Arctic region during the fertile phase [39]. Higher concentrations of fucoidans were found in fertile tissues of five different brown algae when compared to vegetative blades [37]. This suggests that the reproductive phase is a good timing for the sampling of *F. vesiculosus*, if the objective is to obtain or produce biomass for fucoidan extraction.

Photosynthetic pigment composition and concentrations presented a similar pattern throughout the studied locations. Fucoxanthin concentrations ranged between 0.38 and 0.91 mg g^−1^ dw, representing 65% of total carotenoids extracted from *F. vesiculosus*. These values are in the upper range of fucoxanthin content reported for brown seaweeds [40]. In a comparison of fucoxanthin content of ten brown seaweed species, Shannon and Abu-Ghannam [41] found the second highest concentration of this pigment in *F. vesiculosus* (0.70 mg g^−1^ dw), only surpassed by the content in *Alaria esculenta* (0.87 mg g^−1^). Lower concentrations of fucoxanthin from *F. vesiculosus* (0.12–0.15 mg g^−1^ dw) were previously reported [42]. It is important to note that fucoxanthin concentrations are highly dependent on the extraction protocols employed, namely solvent type, temperature, and duration of extraction [41]. These parameters may most likely explain the differences reported in the existing literature on this topic. Furthermore, the concentrations of light-harvesting pigments, such as fucoxanthin, depend on the light history of the macroalgae, with higher concentrations being expected to occur in algae originating from lower light environments. Concentrations of fucoxanthin in *F. vesiculosus* from deeper waters and shaded habitats were found to be higher by factors of 1.6 to 3.2, when compared to specimens from the surface and sunnier habitats [43]. Hence, if not significantly hampering growth, the shading of culture tanks can be an effective measure to produce *F. vesiculosus* or other brown algae in aquaculture if aiming to enhance the yields of fucoxanthin extraction.

## 5. Conclusions

Photosynthetic performance, pigments, and carbohydrate concentrations in *F. vesiculosus* were very similar in different locations along the Ílhavo Channel in Ria de Aveiro, Portugal. This finding indicates a strong resistance by this brown seaweed to changes in abiotic factors, such as salinity and exposure to desiccation. This feature, along with high concentrations of relevant bioactive compounds (e.g., fucoxanthin and fucoidans), make the macroalga *F. vesiculosus* a valuable resource for local aquaculture companies. This macroalga grows prolifically in Ria de Aveiro and can be harvested without damaging the holdfast, thus allowing for continuous regrowth. However, potential contaminant accumulation (e.g., metals) should be monitored to ensure quality control in aquaculture systems. Future research may include the optimization of growth conditions in land-based controlled production systems, the improvement of extraction protocols, the elucidation of chemical structures, and the assessment of bioactivities.

## Figures and Tables

**Figure 1 plants-12-01324-f001:**
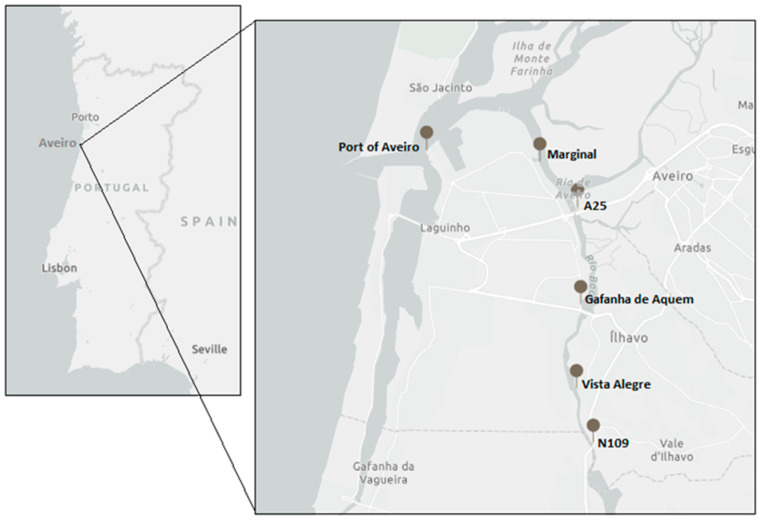
Sampling locations for *Fucus vesiculosus*, from the inlet of Ria de Aveiro and along the Ílhavo Channel, Portugal: Port of Aveiro, Marginal, A25, Gafanha de Aquém, Vista Alegre, and N109. Map generated with ArcGIS Online^®^.

**Figure 2 plants-12-01324-f002:**
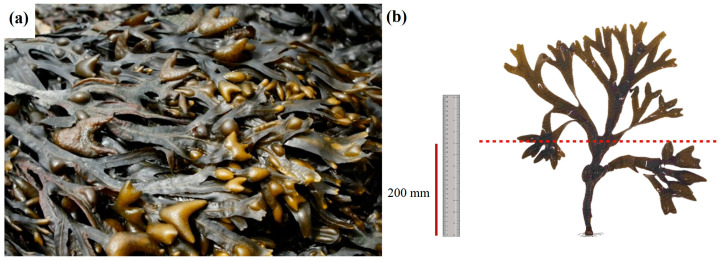
(**a**) The macroalga *Fucus vesiculosus* from Ria de Aveiro; (**b**) sample collection cut 200 mm above the fixation point.

**Figure 3 plants-12-01324-f003:**
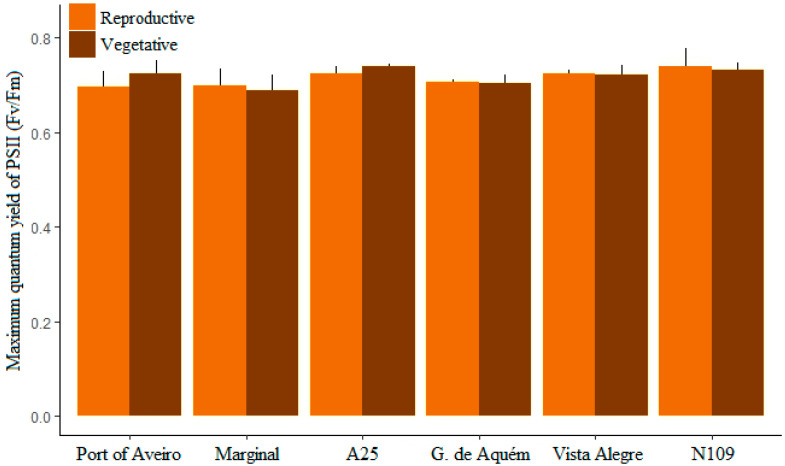
Maximum quantum yield of photosystem II (Fv/Fm) in vegetative and reproductive tissues of *Fucus vesiculosus* at the six locations from the inlet of Ria de Aveiro and along the Ílhavo Channel, Portugal. Mean + standard deviation; *n* = 5.

**Figure 4 plants-12-01324-f004:**
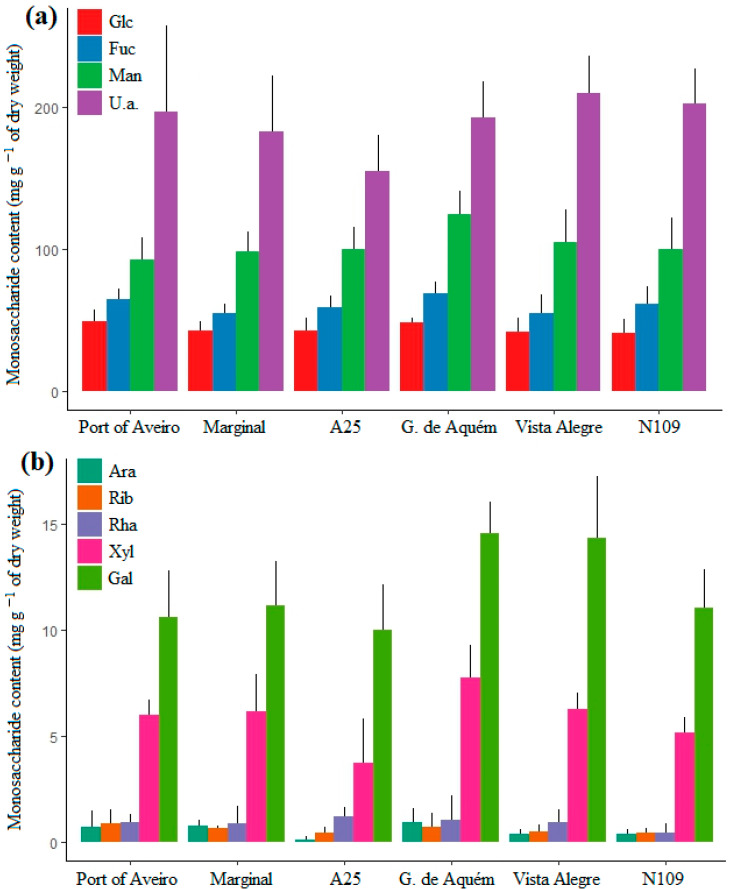
Neutral sugar and uronic acids concentrations of *Fucus vesiculosus* (mg g^−1^ of dry weight; mean + standard deviation; *n* = 5) at the six locations from the inlet of Ria de Aveiro and along the Ílhavo Channel, Portugal. (**a**): the most abundant monosaccharides glucose (Glc), fucose (Fuc), mannose (Man), and uronic acids (U.a.); (**b**): the less abundant monosaccharides, arabinose (Ara), ribose (Rib), rhamnose (Rha), xylose (Xyl), and galactose (Gal).

**Figure 5 plants-12-01324-f005:**
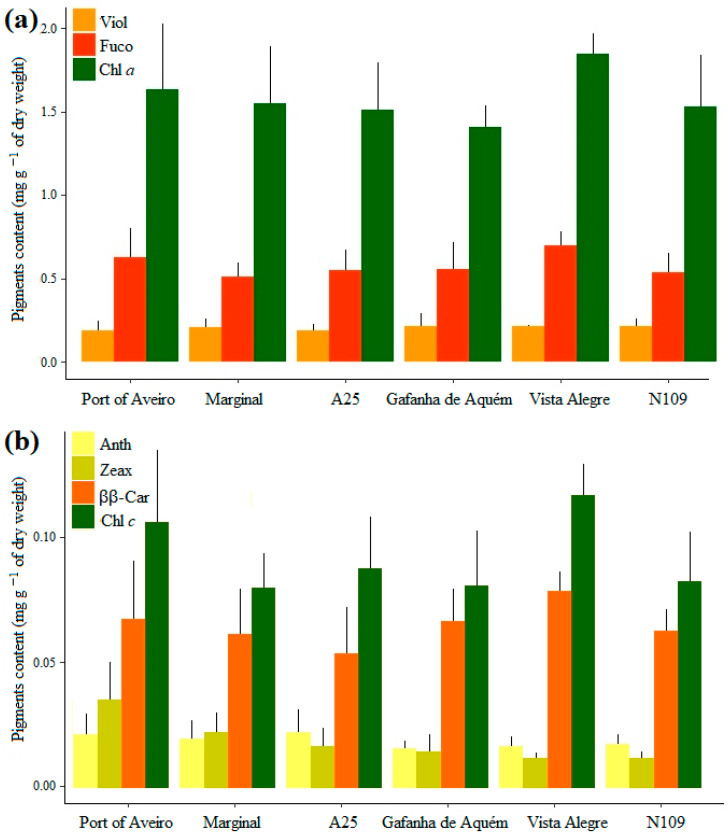
Concentration of photosynthetic pigments of *Fucus vesiculosus* (mg g^−1^ of dry weight; mean + standard deviation; *n* = 5) at the six locations from the inlet of Ria de Aveiro and along the Ílhavo Channel, Portugal. (**a**): The most abundant pigments, violaxanthin (Viol), fucoxanthin (Fuco) and chlorophyll *a* (Chl *a*); and (**b**): the less abundant pigments, antheraxanthin (Anth), zeaxanthin (Zeax), chlorophyll *c* (Chl *c*) and β,β-carotene (ββ-Car).

## Data Availability

The data presented in this study are available within the article and in Appendix A.

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
