# Peer review of "Photosynthetic Pigment and Carbohydrate Profiling of Fucus vesiculosus from an Iberian Coastal Lagoon"

_plants, 2023, doi:10.3390/plants12061324_

Round 1
Reviewer 1 Report
The presented results showed that Fucus vesiculosus from the Ria de Aveiro is a valuable macroalgae resource with significant potential to yield high-value bioactive compounds. The manuscript contains many interesting results that can be applied to the selection of conditions for the cultivation/growth of F. vesiculosus, especially in Portugal. The finding presentation is systematic and transparent. I recommend the manuscript publication in “Plants” journal.
Author Response
Thank you for the positive feedback.
Reviewer 2 Report
The study by Ana C. R. Resende al. entitled “Photosynthetic pigment and carbohydrate profiling of Fucus vesiculosus from an Iberian coastal lagoon” has been reviewed. The manuscript is quite well written, and the rationale is clear, however, the research is weak end presents some flaws and for these reasons, the work cannot be published in its present form. Some aspects of the work need improvement.
Major:
1- Please add a full characterization of the sampling sites such as coordinates, mean water temperature, range of salinity, exposure time to desiccation (you based the discussion on some of them), including every kind of close anthropogenic activity, etc…
2-Please add some further basic analysis to support your hypothesis about F. vesiculosus as a valuable resource for local aquaculture companies, such as ash content, minerals concentrations,
3- Please evaluate the possible accumulation of toxic metals in your samples
Minor:
Please add a more detailed description of both the species studied and the genus it belongs to (habitat, characteristics, farming, etc..., including traditional uses).
I would urge the authors to accommodate the suggestions reported above, to improve the quality of the manuscript.
Author Response
As referred in the manuscript, this is a preliminary study, a snapshot of the biochemical profile of Fucus vesiculosus in the Ria de Aveiro coastal lagoon. We focused in the high-value compounds identified by the aquaculture company that participated in this study (A4F- Algae for future LDA): fucoxanthin and fucoidans. As so, the paper is submitted as a Communication. Nevertheless, I believe the data obtained shows that F. vesiculosus is a valuable resource and the next step is to culture the macroalga under controlled land-based systems. In the Conclusions, we have included the need to screen for potential contaminants (e.g., metals) in aquaculture systems, mentioned by the Reviewer. We have included further information in the paper on the sampling sites such as coordinates and near anthropogenic activity (Materials & Methods) and a description of the species (Introduction), as suggested by the reviewer.
Introduction:
“The bladder wrack Fucus vesiculosus Linnaeus, 1753 is a large brown seaweed characterized by its frond with prominent midrib; oval to spherical air bladders, usually paired; and dichotomously branched. This species is similar to Fucus spiralis Linnaeus 1753 with which it hybridizes. Fucus vesiculosus has a wide distribution, dominating specific shallow intertidal macroalgal communities on rocky shores. It is the most studied species of the genus Fucus with several benefits to human health, namely suppressing mineral deficiencies, displaying high iodine content and bioactive compounds such as fucoidans and fucoxanthin”.
Materials & Methods:
“Fucus vesiculosus (Ochrophyta, Phaeophyceae) was collected from six locations from the inlet of Ria de Aveiro and along the Ílhavo channel Port of Aveiro (40°38'49.7"N 8°44'04.2"W), Marginal (40°38'42.0"N 8°41'52.3"W), A25 (40°37'57.2"N 8°41'06.5"W), Gafanha de Aquém (40°36'31.4"N 8°41'02.3"W), Vista Alegre (40°35'16.5"N 8°41'06.7"W), and N109 (40°34'27.1"N 8°40'47.7"W).”
“The Ílhavo channel has around 15 km of extension and ranges from 60 to 500 meters in width. It is considered the smaller and narrower channel of the Ria de Aveiro lagoon, receiving the freshwater input from the Boco river [21]. The salinity and the water temperature range from 0 to 38, and from 12 to 23 °C, respectively, depending on the season and distance from the mouth [21,22]. The sampling locations were chosen accordingly to a known salinity and tidal gradient: salinity increases from N109 towards Port of Aveiro, while exposure to desiccation periods increases in the inverse direction [21,22]. All samples were collected in a rocky substrate in dense populations, except for Vista Alegre that presents muddy substrate and a low-density population. Port of Aveiro, Marginal and A25 locations are more influenced by anthropogenic activities, such as those occurring in the main commercial port of Aveiro and the industrial fishing port.”
Conclusions:
“However, potential contaminant accumulation (e.g., metals) should be monitored to ensure quality control in aquaculture systems [44].”
Reviewer 3 Report
This manuscript describes the carbohydrate and photosynthetic pigment composition, and the photosynthetic efficiency of different tissues from the brown algae Fucus vesiculosus. Samples were collected in six different locations along the Ria de Aveiro (Portugal), were there is a gradient in water salinity and in the frequency of desiccation exposure. The objective was to characterize biochemical profiles of natural populations, as a first approximation to optimize sustainable aquaculture of this brown seaweed.
Contrary to what was expected, all populations showed similar compositions in neutral and acid sugars, photosynthetic pigments and photosynthetic efficiency, pointing towards a negligible effect of water salinity and desiccation stress on those parameters. At least, in that particular sampling time (summer-dry season).
The paper is well written, and the analytical techniques and statistical analysis were correctly performed. However results are merely descriptive.
Minor
Page 6, line 183
“Total pigments concentration ranged between 1.8 and 3.8 mg g-1 dw” Are those figures correct? It seems, from the mean and standard deviation values informed for the different photosynthetic pigments, that the lower value is out of range
Author Response
The values presented are correct. The raw data can be accessed in the submitted Excel file as supplementary material.
Round 2
Reviewer 2 Report
All the issues raised by this reviewer were adequately answered. This manuscript is now ready for possible publication in Plants.